# The Role of Regulated Programmed Cell Death in Osteoarthritis: From Pathogenesis to Therapy

**DOI:** 10.3390/ijms24065364

**Published:** 2023-03-10

**Authors:** Suqing Liu, Yurong Pan, Ting Li, Mi Zou, Wenji Liu, Qingqing Li, Huan Wan, Jie Peng, Liang Hao

**Affiliations:** 1Department of Orthopedics, Second Affifiliated Hospital of Nanchang University, Nanchang 330006, China; 2Queen Marry College, Nanchang University, Nanchang 330006, China; 3The Second Clinical Medical College, Nanchang University, Nanchang 330006, China

**Keywords:** programmed cell death, osteoarthritis, pathogenesis, therapy

## Abstract

Osteoarthritis (OA) is a worldwide chronic disease that can cause severe inflammation to damage the surrounding tissue and cartilage. There are many different factors that can lead to osteoarthritis, but abnormally progressed programmed cell death is one of the most important risk factors that can induce osteoarthritis. Prior studies have demonstrated that programmed cell death, including apoptosis, pyroptosis, necroptosis, ferroptosis, autophagy, and cuproptosis, has a great connection with osteoarthritis. In this paper, we review the role of different types of programmed cell death in the generation and development of OA and how the different signal pathways modulate the different cell death to regulate the development of OA. Additionally, this review provides new insights into the radical treatment of osteoarthritis rather than conservative treatment, such as anti-inflammation drugs or surgical operation.

## 1. Introduction

Osteoarthritis is a chronic disease [1] involving inflammatory reactions [2] that mainly damage the surrounding tissue and cartilage [3]. The effect of osteoarthritis on cartilage leads to many other diseases as well as increases the severity of osteoarthritis itself [4]. The most common symptom of osteoarthritis is pain; in the early stages of the disease, pain is felt mainly upon exercise, while later stages are characterized by pressing pain and continuous pain [5]. In addition, osteoarthritis causes joint stiffness and loss of motor function [6], resulting in loss of confidence and self-abasement [6,7]. There are many risk factors for osteoarthritis, including dyslipidemia and type Ⅱ diabetes [8], high-intensity joint impact [4], lack of antioxidants [4], high bone mineral density, and high age [9]. Recently, several studies on the correlation between programmed cell death and osteoarthritis have revealed that programmed cell death may be an important risk factor for osteoarthritis.

Programmed cell death (PCD) is an active mode of cell death that is regulated by various genes. All types of programmed cell death play an important role in the development of an organism, and uncontrolled programmed cell death can cause developmental damage. For example, uncontrolled programmed cell death can cause neurological diseases [10], such as Parkinson’s disease and Alzheimer’s disease, as well as cancer. Uncontrolled programmed cell death is also a risk factor for secondary osteoarthritis.

Programmed cell death may be lytic or non-lytic [11], The main type of non-lytic cell death, wherein cell contents don’t overflow upon cell death, is apoptosis. In apoptosis, cell growth and development stop, and a cell undergoes a controlled death [12]. There are two main pathways that mediate apoptotic cell death, intrinsic and extrinsic pathways. In many osteoarthritis cases, apoptosis is found to occur in the arthrodial cartilage [13]. Pyroptosis is another important type of programmed cell death [14], which causes membrane perforation, nuclear condensation, and chromosomal lysis [15]. Pyroptosis in osteoblasts can cause loss of arthrodial cartilage, leading to osteoarthritis. Ferroptosis is a type of programmed cell death in which iron and reactive oxygen species (ROS) cause structural changes in mitochondria, such as cleavage of the outer mitochondrial membrane and degeneration of mitochondrial cristae [16,17,18,19,20,21]. Although there are only a few studies on the relationship between ferroptosis and OA, it has been shown that, in cartilage cells, ferrostatin-1 can decrease the activity of ferroptosis-related proteins [22]. Unlike necrosis, which is caused by wounds, necroptosis is a type of caspase-independent programmed cell death [23]. Autophagy eliminates internal aggregates of damaged cells, such as aggregates of misfolded proteins, as well as senescent or damaged organelles [24]. The process of autophagy starts with the formation of an isolation membrane by the endoplasmic reticulum and the trans-Golgi network [25]. This isolation membrane then wraps around the protein aggregates and damaged and senescent organelles, enabling their fusion with lysosomes [26]. The lysosomal enzymes can then break these components down so that they can be reused to produce new proteins and organelles; autophagy is thus called a “recycling factory” [27,28,29]. In general, different types of programmed cell death play different roles in OA. Most of these, such as apoptosis, pyroptosis, cuproptosis, ferroptosis, and necroptosis, can induce or exacerbate OA. Autophagy can inhibit the exacerbation of OA and, thereby, protect cartilage. Due to these functions of PCD, there has been research into the discovery of drugs targeting signal pathways involved in various types of PCD in order to treat or relieve the symptoms of OA.

There are currently many treatment options for osteoarthritis. The main therapeutic prescription for OA involves conservative treatments to reduce symptoms. However, this treatment strategy does not treat OA at the source. Therefore, drugs targeting programmed cell death, such as anti-cytokine drugs, can be a good option for treating the OA pathology [30].

## 2. Mechanisms of Cell Death in Osteoarthritis

### 2.1. Apoptosis

Lacunar emptying and reduced cell density in osteoarthritic cartilage were discovered through histological analysis, suggesting that cell death may occur in this cartilage throughout the duration of OA and may even contribute to OA development [31,32]. A positive correlation was found between the number of apoptotic chondrocytes and the extent of OA-mediated degradation in the human cartilage [33]. The apoptosis of chondrocytes and inflammatory responses are important factors in the pathogenesis of OA, with osteoarthritic chondrocytes releasing inflammatory mediators, including, Interleukin-1β(IL-1β), Tumor necrosis factor (TNF), prostaglandins, and nitric oxide (NO) [34,35].

Chondrocyte destruction, involving inflammation and apoptosis, is facilitated by mitochondrial malfunctioning brought on by excessive oxidative stress, abnormalities in the respiratory chain, and an imbalance in the mitochondrial dynamics [36]. Excess superoxide anions, hydrogen peroxide (H_2_O_2_), and NO, resulting from elevated oxidative stress due to intrinsic apoptosis, are crucial in the pathogenesis of OA [37,38].

Immunostaining of human OA cartilage with anti-nitrotyrosine antibodies revealed that the inducible nitric oxide synthase (iNOS) enzyme is upregulated in OA chondrocytes, resulting in the excess production of NO in the cartilage [39,40]. Both endogenous and exogenous NO can induce apoptosis through a mitochondria-dependent mechanism [41,42], involving the obstruction of mitochondrial respiration and production of cytochrome C (Cyt C) and caspase-9 [43]. The phosphate-dependent apoptosis of chondrocytes is also mediated by NO [44]. The inner mitochondrial membrane contains a large conductance channel called PTP [45]. Both high calcium and ROS levels can cause PTP to open, inducing mitochondrial permeability transition pore (MMP) loss [46,47]. It was found that mitochondrial dysfunction, including increased mitochondrial membrane permeability and decreased MMP expression, can promote cytochrome C (Cyt-C) migration from the mitochondrial matrix to the cytoplasm [48], triggering apoptosis through caspase activation and raising the BAX/Bcl-2 ratio [49]. One study showed that, due to the increased ROS burden of aged chondrocytes, OA patients had greater mtDNA damage than individuals with normal chondrocytes [50,51,52]. *STING* expression was found to be significantly increased in both human and mouse OA tissues and chondrocytes exposed to IL-1β. *STING* overexpression increased apoptosis in both chondrocytes treated and untreated with IL-1β [53].

As an inflammatory factor, IL-1β plays an extremely important role in the development of osteoarthritis [54,55,56]. In the OA tissues of both humans and mice, as well as in chondrocytes exposed to IL-1β, the expression of *STING* was noticeably increased. Moreover, an increase in the expression of *STING* could induce apoptosis by inducing extracellular matrix (ECM) degradation [57].

TRAIL, which can bind to death receptors exposed on the cell surface to trigger apoptosis, can directly cause chondrocyte apoptosis via the extrinsic pathway; in an experimental OA rat model, TRAIL and DR4 were found to be overexpressed in the cartilage [58,59]. These findings indicated that trail-induced chondrocyte apoptosis influences OA pathogenesis.

Many non-coding RNAs, including long-noncoding RNAs and circular RNAs, can regulate chondrocyte apoptosis through different mechanisms. Some of the mechanisms that can promote chondrocyte apoptosis involve Circ_0136474 [60,61], long non-coding RNA plasmacytoma variant translocation 1 [62], LINC00707 [63], and CircZNF652E [64]; the mechanisms that can prevent chondrocyte apoptosis include circANKRD36 [65], long noncoding RNA NAV2-AS5 [66], and circ_0020014 [67]. Circular RNAs act as sponges for miRNAs to suppress their activity and, thereby, regulate the expression of their downstream target genes [68]. The OA cartilage expresses miR-146a upon being triggered by a variety of microbes and pro-inflammatory cytokines, including IFN-α and IL-1β [69,70]. For instance, miR-146a-5p expression significantly increased caspase-3 activation, PARP degradation, and Bax expression while repressing Bcl-2 expression in human OA tissues, thereby accelerating the chondrocyte apoptosis [69]. MiR-146a could increase human chondrocyte apoptosis by specifically suppressing Smad4 expression in mechanically damaged cartilage [71].

### 2.2. Pyroptosis

Pyroptosis is a type of lytic cell death that has many characteristics in common with and many characteristics different from apoptosis. During pyroptosis, the cell first expands and then generates gigantic bubbles from the plasma membrane [72,73]. Pyroptosis is primarily caused by two pathways: the canonical inflammasome pathway, which is mediated by caspase-1, and the non-canonical pathway, which is mediated by caspases 4, 5, and 11 [74]. In the canonical inflammasome pathway, the NLR family pyrin domain containing 3 (NLRP3) inflammasome induces pyroptotic cell death through Gasdermin D (GSDMD), the release of which is dependent on caspase-1 [75]. Many risk factors for osteoarthritis, such as obesity, basic calcium phosphate (BCP), and aging, can activate NLRP3 inflammasome assembly and, thus, induce OA [76]. The level of pyroptosis-associated inflammasomes is high in the articular fluid of patients with OA and can increase the levels of IL-1β and IL-18, both of which can increase cartilage cell pyroptosis and inflammatory responses in osteoarthritis [76]. The pro-inflammatory cytokines IL-1β and IL-18 also play a role in the induction of pain in OA and hyperalgesia. Some inhibitors of NLRP3 inflammasomes, such as CY-09, can significantly inhibit OA deterioration and alleviate the OA development [77]. P2X7 purinergic receptor (P2X7R) can also aid NLRP3 inflammasome assembly and, thereby, cause pyroptosis [78]. Furthermore, pyroptosis not only directly leads to OA but also interacts with other kinds of PCD, such as apoptosis, to promote osteoarthritis. For example, excessive production of apoptotic bodies and increased calcification of cartilage tissue can induce pyroptosis and, thereby, exacerbate OA [79]. During OA development, pyroptosis can cause pathological changes in joints affected by inflammation, such as cartilaginous defects, cartilage fracture, and arthromeningitis [80].

### 2.3. Necroptosis

Necroptosis, which is mediated by Receptor-interacting serine/threonine-protein kinase 1(RIPK1) and Receptor-interacting serine/threonine-protein kinase 3(RIPK3), has a mechanism similar to that of apoptosis and a morphology similar to that of necrosis [81,82]. A well-known necroptosis-inducing pathway is Tumor necrosis factor receptor 1(TNFR1) signaling through the complex IIb [83]. Complex IIa can be transformed into complex IIb, which is made up of RIPK1, RIPK3, FADD, and caspase-8; when CASP8 is blocked, complex IIb can develop into a necrosome [84]. The presence of necroptotic chondrocytes in fractured human and mouse cartilages was revealed by immunohistochemical staining for markers of necroptosis, including RIP3, p-MLKL, and MLKL [85]. According to earlier studies, the processes of necroptosis and cartilage destruction may be related to oxidative stress and cytokine generation in OA [86,87,88]; RIPK plays a part in all forms of osteoarthritis. Oxidative stress is a key mediator of programmed necrosis in temporomandibular joint osteoarthritis and has a significant impact on the disease. Cartilage breakdown was found to be accelerated by TNF- and RIPK1/RIPK3-mediated programmed necrosis. The programmed necrosis pathway was strengthened when apoptosis was suppressed [89]. However, the TRIM24-RIP3 axis also allows RIP3 to function in cartilage in an MLKL-independent manner [85]. Another study hypothesized that the TLR-TRIF-RIP3-IL-1β axis enhances arthritis development independently of MLKL [90].

As a primary regulator of cell death, in the presence of appropriate downstream signals, RIP1 promotes not only necroptosis but also apoptosis, unlike RIP3, which predominantly mediates necroptosis [91]. A study by Cheng recently suggested that RIP1 expression is considerably increased in the cartilage of OA patients and OA rat models and provided in vivo evidence that overexpression of intra-articular RIP1 is enough to trigger rat OA symptoms; these findings emphasized the critical function of RIP1 in OA development through the regulation of chondrocyte necroptosis and ECM breakdown. Interestingly, bone morphogenetic protein 7 (BMP7) was discovered as a new downstream target of RIP1 in chondrocytes, revealing a non-canonical necroptosis regulation method; moreover, it was found that MLKL is not necessary for OA pathogenesis and chondrocyte necroptosis induced by RIP1 [92]. However, the physiological and pathological roles of RIP3 and RIPK1 in cartilage are yet to be investigated, and it is unknown whether RIP3-mediated or RIPK1-mediated necroptosis regulation is involved in OA etiology [92,93].

### 2.4. Ferroptosis

Ferroptosis is a kind of iron-dependent programmed cell death that mainly affects the structures of cellular organelles, especially the mitochondrial structure, and is characterized by the accumulation of lipid peroxides [94]. In ferroptosis, iron accumulation can increase ROS production [95]. These ROS can then induce the peroxidation of lipids on the outer mitochondrial membrane, leading to cell death [96]. In vitro control experiments revealed that the iron ion concentration in the cartilage synovial fluid was significantly higher, and the glutathione peroxidase 4 (GPX 4) content in the cartilage was lower in the osteoarthritis group compared to that in the normal group [96]. GPX4 was found to protect chondrocytes by inhibiting the OA-related ferroptosis [97]. The inactivity of the lipid repair enzyme GPX4 can lead to an increase in the lipid-based reactive oxygen species content and, thus, induce ferroptosis [4]. In ferroptosis, iron plays a catalytic role. For example, iron can catalyze LOXs, which are important for phospholipid peroxidation-related metabolism [98]. Moreover, ferroptosis can increase the expression of MMP13 and decrease the expression of collagen Ⅱ [94]. Matrix metalloproteinase 13 (MMP13) is an important enzyme involved in arthrodial cartilage degeneration, and cartilage ECM degeneration is a hallmark of osteoarthritis [99]. Through clinical experiments, Yao et al. showed that ferrostatin-1 could decrease the expression of ferroptosis-related proteins to inhibit ferroptosis and thereby relieve OA symptoms [22].

### 2.5. Autophagy

Autophagy is another type of programmed cell death that plays an important role in OA. The main regulatory factor of autophagy is mTOR. mTOR can inhibit autophagy by phosphorylating ULK1, which inhibits the function of AMPK in the ULK1 [100]. The main function of AMPK in ULK1 is that the AMPK can phosphorylate ULK1. Using cell autophagy, phosphatidylinositol 3-kinase catalytic subunit type 3 (PlK3C3), the mammalian ortholog of yeast Vps34, can phosphorylate PIP2 to PI3P, which is necessary for recruiting autophagic vacuoles [101]. Autophagy is important for OA development. First, chondrocyte autophagy can regulate various chondrocyte activities [102]. Moreover, autophagy can break down misfolded proteins and damaged organelles in cartilage cells for recycling and, thereby, mitigate the development of osteoarthritis [103]. Although mTOR signaling pathway activation is essential for cartilage development, in a mouse model of OA, mTOR signaling pathway overexpression in autophagic cells could induce OA. One study showed that the activation of AMPK by metformin could relieve OA symptoms, as AMPK can inhibit mTORC1 activity. Moreover, their overall effect can induce autophagy to inhibit the senescence of the chondrocytes and the development of OA [104]. Autophagy can be used to break down the source of intracellular ROS. An important step in OA development is ROS, acting as a second message to induce mitochondrial damage and ER activation [105]. If the autophagy of cells with mitochondrial damage (mitophagy) does not occur, excess ROS are released from them and induce abnormal cell death, leading to OA [106]. Many risk factors for OA, such as high-intensity impact, inflammatory cytokines, and high age, can increase ROS in the chondrocytes [107]. Moreover, autophagy is important for mesenchymal stem cells, and mesenchymal stem cells are crucial for the stabilization and repair of cartilage tissue [101].

### 2.6. Cuproptosis

Copper and other trace elements are essential for the body. It is critical for these metals to be present in sufficient quantities in cells [108]. Dysregulation of the intracellular bioavailability of copper, a critical cofactor, can cause oxidative stress and cytotoxicity [109]. Excess copper, for example, disturbs the iron–sulfur cofactors and promotes the formation of hazardous reactive oxygen species via accelerating Fenton reactions [110]. The known cellular mechanisms and enzymatic targets of Cu, however, fall short of adequately explaining the cellular response to the Cu toxicity [108]. The fundamental cause of copper-related cell death is the intracellular buildup of copper ions, which bind to and aggregate the fatty acylated proteins involved in the TCA cycle. Blocking of the TCA cycle by Cu ions causes protein toxic stress, which results in cell death [111]. Further research is necessary to properly understand the precise mechanisms underlying cuproptosis. *SLC31A1*, *PDHB*, *PDHA1*, *LIPT1*, *LIAS*, *DLD*, *FDX1*, *DLST*, *DLAT*, and *DBT* are the top 10 cuprotosis genes linked to immune infiltration in osteoarthritis. Results of this study involving a risk model showed that the copper cell death gene *PDHB* might be a risk factor for osteoarthritis. Two E1 isoforms of the pyruvate dehydrogenase complex, PDHB, and PDHA1, are mostly present in cellular mitochondria and catalyze the conversion of glucose-derived pyruvate to acetyl-coA [112,113,114]. A mouse study of osteoarthritis also revealed that high acetyl-coA buildup, induced by matrix metalloproteinases, led to significant cartilage degeneration and chondrocyte apoptosis [115]. LIPT1 and LIAS are both involved in the production of mitochondrial fatty acids [116]. According to one study, the role of free fatty acids in subchondral bone injury may be more dependent on the inflammatory response and immune system than on related signaling pathways [117]. Based on the findings of the abovementioned studies, it can be hypothesized that copper cell death genes are crucial for immune infiltration in osteoarthritis; however, there have been few investigations into the involvement of copper cell death-related genes in the immune regulation of osteoarthritis. Antigen-presenting cells, mast cells, dendritic cells, and chemokine receptor 2 (CCR2) may have significant effects on the regulation of copper cell death genes in osteoarthritis [118,119,120,121] (Figure 1).

## 3. Modulation of Cell Death in Osteoarthritis

### 3.1. NF-κB Signaling Pathway

Mechanical stresses and elevated levels of inflammatory cytokines, such as TNF-α, IL-1β, and IL-6, in the joints of patients with OA play essential roles in damaging the cartilage homeostasis [122]. Nuclear factor kappa-B (NF-κB) signaling can regulate oxidative stress-activated mechanical and inflammatory processes to contribute to the damage of cartilage tissue [123]. The human NF-κB protein family is made up of five members. These share an N-terminal DNA-binding/dimerization domain (Rel homology domain, RHD), through which they bind as homodimers or heterodimers to 10-base pair κB sites on DNA [124,125]. There are two main NF-κB signaling pathways, the canonical pathway and the noncanonical pathway. Most physiological NF-κB stimuli, such as TNF-α, LPS, IL-1β, and ROS, can induce the canonical pathway [126]. In unstimulated cells, NF-κB dimers interact with inhibitory IκB proteins, which sequester inactive NF-κB complexes in the cytoplasm [127]. Under stimulated conditions, upstream receptors activate and phosphorylate an IKK complex consisting of IKKα, IKKβ, and IKKγ (also known as NEMO), which then phosphorylates IκB proteins. Phosphorylated IκB proteins are then ubiquitinated and degraded by the proteasome, leading to the release and nuclear translocation of NF-κB dimers [128,129]. Only a small number of TNF superfamily receptors can activate the non-canonical NF-κB pathway [130]. The NF-κB-inducing kinase is the key kinase in this pathway. Upon exposure to stimuli, the degradation of NIK induced by TRAF3 and 2 and clAPs1 and 2 will be prevented [131,132]. This allows NIK, together with IKKα, to phosphorylate NF-κB2 (p100); p100 is then processed into p52 by proteasomes, releasing the RelB/p52 complex and mediating the persistent activation of the RelB/p52 complex [133,134,135].

The relationship between NF-κB signaling-mediated cartilage cell apoptosis and OA exacerbation has been shown by various studies [136,137]. NF-κB bidirectionally regulates chondrocyte homeostasis. The important subunit RelA can induce the expression of anti-apoptotic genes, such as pik3r1, to protect chondrocytes against apoptosis [138]. Inhibition of miR-495 suppresses chondrocyte apoptosis and promotes chondrocyte proliferation by activating the NF-κB signaling pathway through the up-regulation of CCL4 in OA. Moreover, NF-κB can induce the expression of anti-apoptotic genes to inhibit TNF-α-induced cell death, but it can also activate the TNF-α-induced cell death [139,140]. Several NF-κB-activated factors, such as SAM68, TCF4, and RIPK, can enhance the chondrocyte apoptosis [122]. In addition, the NF-κB target gene HIF-2α can accelerate Fas-mediated chondrocyte apoptosis in OA cartilage to worsen OA [141,142]. The NF-κB signal pathway can also induce the expression of TLR-2 and INOS, both of which can regulate the apoptosis of cartilage cells [143,144]. NF-κB can also induce the generation of PGE2 to cause apoptosis of cartilage cells [145]. The combination of ROS and RAGE can lead to apoptosis of cartilage cells through NF-κB [146]. In summary, these findings indicate that NF-κB signaling can be a useful drug target for the treatment of OA.

### 3.2. mTOR Signaling Pathway

The mammalian target of rapamycin (mTOR) is an important signal sensor in autophagy. mTOR is essentially a serine/threonine kinase. mTOR forms two complexes, mTORC1 and mTORC2. Since the two complexes differ in composition, they play different roles in the mTOR signaling pathway [147]. The mTOR signaling pathway is an important pathway that can promote the synthesis of proteins, lipids, nucleotides, and other nutrients or attenuate the process of autophagy [148]. mTOR has many upstream pathways, of which the PI3K/AKT/mTOR is the most important for OA, especially the signaling pathway induced by class I PI3Ks. The PI3K/AKT/mTOR pathway can activate mTORC1 to inhibit autophagy, increase apoptosis, and, thereby, exacerbate OA [149]. There are many kinds of molecules that can bind to GPCRs or activate receptor tyrosine kinases (RTKs) to activate phosphoinositide 3-kinase (PI3K). Activated PI3K can activate its downstream factor AKT through phospholipids produced through PI3K [150]. PI3K can also change PIP2 to PIP3 in order to activate AKT [151]. AKT can then directly activate mTORC1 by phosphorylating it or indirectly activate mTORC1 by phosphorylating TSC2 [152]. Tuberous sclerosis complex (TSC) is another upstream target of mTOR; the function of TSC is to inhibit mTORC1. TCS can change Rheb from its GTP-binding form to its GDP-binding form. Rheb can only activate mTOR in its GTP-binding form [153]. Besides PI3K and TSC, the AMPK/mTOR pathway also acts as an important upstream pathway of mTOR in case of insufficient energy supply (Amp/ATP higher). The γ-subunit of adenosine monophosphate-activated protein kinase (AMPK) can bind to AMP or ADP [154] to influence the α-subunit [155], which activates AMPK; activated AMPK can then inhibit the activity of mTOR [156].

mTOR has many downstream processes. It can regulate cellular catabolism by inhibiting autophagy in multiple ways. Under nutrient-rich conditions, mTORC1 can phosphorylate and inhibit the binding of ULK1 to AMPK as well as phosphorylate Atg13; both these actions can inhibit the activity of ULK1 [148]. ULK1 is an important kinase core complex for autophagy. After receiving a stimulus, Atg13 anchors ULK1 to PAS, following which almost all kinds of Atg proteins are recruited to PAS, inducing cell autophagy [157]. ULK1 can directly phosphorylate AMBRA1, enabling the binding of Beclin-1 and VPS34 to promote autophagy [158]. In some special pathways, mTORC1 can phosphorylate the Ser498 site of *UVRAG* genes and phosphorylate the Ser157 site in Pacer to inhibit autophagy [159,160].

The PI3K/AKT/mTOR signaling pathway acts on some factors involved in the development of OA. For example, cartilage homeostasis is highly related to this signaling pathway [149] because the PI3K/AKT pathway can help in the synthesis of the ECM [161]. The PI3K/AKT pathway is also related to the inflammatory response induced by IL-1β [162]. Furthermore, the PI3K/AKT pathway can regulate the development of OA by regulating programmed cell death, for example, by inhibiting the apoptosis of chondrocytes.

### 3.3. JAK Signaling Pathway

The Janus Kinase/signal transducer and activator of transcription (JAK/STAT) signal pathway can be activated by many pro-inflammatory factors that play key roles in OA, such as IL-1β, IL-6, and TNF-α [163]. The JAK/STAT pathway plays important roles in many cellular activities, such as the formation of skeletal muscle and the transformation of normal cells into cancerous cells [164,165,166]. In the canonical JAK/STAT signaling pathway, cytokines bind to the receptors and dimerize the receptors to recruit JAK, following which, JAK promotes the phosphorylation of STAT [167,168]. Phosphorylated STAT proteins then bind to form STAT-STAT dimers, which can bind to DNA and regulate gene expression [165]. The STAT-STAT dimers can then be deactivated through the dephosphorylation [169]. In the noncanonical JAK/STAT signaling pathway, most STAT proteins localize to the mitochondria, and STAT3 localizes to the ER to attenuate the apoptosis [170]. JAK can also be activated by the tumorigenic tyrosine kinase [171]. In OA, pro-inflammatory factors act as stimulators; for example, TNF-α can positively regulate the JAK/STAT signal pathway to increase the expression of STAT and induce the death of chondrocytes [172]. IL-6 is also important for regulating the JAK/STAT pathway in OA; it can make M1 phenotype macrophages more active, and these macrophages can create an excessive inflammatory response, causing more severe symptoms. In contrast, IL-4 and IL-13 can activate M2 phenotype macrophages through the JAK/STAT3 pathway; these macrophages have anti-inflammatory effects that can reduce the symptoms of OA [169].

### 3.4. HIF Signaling Pathway

Because there are no arteries in articular cartilage, oxygen can only reach chondrocytes passively through synovial fluid and subchondral bone [173]. Chondrocytes need the hypoxia-inducible transcription factors HIF-1 and -2 to adapt to hypoxic environments.

HIF-1 controls autophagy and apoptosis and is necessary for cartilage homeostasis [174]. Transgenic studies showed that conditional HIF-1α KO could result in significant cell death of chondrocytes in the development plate, shedding light on the importance of HIF-1α for chondrocyte survival [175]. Mature chondrocytes in the growth plate survived under an autophagic condition induced by HIF-1α before undergoing apoptotic cell death [176]. Inhibition of mTOR through the regulation of the beclin-1/Bcl-2 complex and activation of the AMPK enzyme may be the processes through which HIF-1α induces autophagy [176,177]. Chen et al. recently showed that Bcl-2 regulation might play a part in the HIF-1α-mediated autophagy [178]. In an HIF-1α-dependent manner, hypoxia elevated the expression of miR-146a; moreover, through lowering the expression of the autophagy inhibitor Bcl-2, miR-146a enhanced the autophagy [179]. According to one study, when the HIF-1 expression level was dramatically suppressed, the expression levels of cleaved caspase-3, Bax, and cyt C were increased, while the expression level of Bcl2 was significantly increased. These findings suggested that the knockdown of HIF-1α increased mitochondrial ROS production as well as hypoxia-induced apoptosis [179]. Through the activation of p38 kinase and PI3K in hypoxia, catabolic factors, such as oxidative stress and IL-1β, may promote HIF-1α expression in chondrocytes. Hypoxic chondrocytes generate energy and survive by expressing HIF-1α, and chondrocytes lacking HIF-1α expression have enhanced IL-1β-induced apoptosis [180]. It is interesting to note that HIF-1α can activate the nucleotide-binding oligomerization domain (NOD)-like receptor (NLR) family pyrin domain-containing 3 (NLRP3) inflammasome, the assembly and activation of which results in an inflammatory cascade response based on the maturation and secretion of IL-1β, IL-18, and HMGB1 and even results in pyroptosis [181,182,183]. Additionally, Zhang et al. [182] found that elevated HIF-1α expression in knee OA worsened synovial fibrosis via fibroblast-like synoviocyte pyroptosis [182]. Chondrocyte survival and death are primarily regulated by an HIF-1α/HIF-2α imbalance. Apoptosis and autophagy result from an imbalance caused by decreased HIF-1α expression and increased HIF-2α expression [178]. OA pathogenesis may involve a switch from HIF-1α to HIF-2α since HIF-1α maintains cartilage, while HIF-2α causes endochondral ossification and cartilage breakdown.

As a powerful negative regulator of the autophagic flow, HIF-2α mediates the death of chondrocyte cells [174,184]. HIF-2α is not only affected by oxygen levels in vitro, but it is also sensitive to non-hypoxic stimuli, such as inflammatory cytokines, growth factors, and catabolic stress. IL-1β and TNF-α were found to promote HIF-2α expression in chondrocytes via the NF-κB and p38 pathways [185,186]. Because IL-1β and TNF-α are putative NF-κB signaling ligands, the NF-κB signaling pathway is an upstream regulator of the HIF-2α [186,187]. Moreover, activating the NF-κB/Hif-2α axis increases cleaved caspase-3 levels and reduces Bcl-2 levels in vivo and in vitro, which is opposite to the effects of HIF-1α activation. This effect of NF-κB/Hif-2α axis activation is linked to SNP-induced apoptosis and matrix breakdown in chondrocytes and articular cartilage [188]. Moreover, HIF-2α downregulates autophagy in maturing chondrocytes by activating Akt-1 and mTOR to facilitate cell death while potentiating Fas-mediated chondrocyte apoptosis [174,189]. Through FACS experiments, it was found that overexpression of HIF-2α elevated Fas mRNA as well as Fas surface expression [142]. A recent study revealed a unique HIF-2α mechanism that functions throughout the course of OA; it promotes cell death via lipid oxidation, ROS buildup, and ferroptosis regulators [190].

### 3.5. URP Signaling Pathway

Unfolded protein response (URP), a process that seeks to restore ER homeostasis, is triggered when an excess of unfolded or misfolded proteins accumulate within the ER lumen [191,192]. In vitro studies have shown that biomechanical injury, nitric oxide, and IL-1 are some components of OA pathogenesis that can activate the UPR in cultured chondrocytes [193,194].

The ER lumen resident chaperone glucose-regulated protein 78/binding immunoglobulin protein (GRP78/BiP) is released from three transmembrane receptors, including inositol-requiring enzyme 1 (IRE-1), protein kinase RNA-like ER kinase (PERK), and activating transcription factor 6 (ATF-6), to start the unfolded protein response (UPR). Three receptor signaling pathways are triggered as a result, with the main objectives of inhibiting protein translation and destroying misfolded proteins [195]. These ER stress sensor pathways are highly active in the cartilage of patients with OA, suggesting that ER stress signaling is sufficient to cause arthritis [196]. The GRP78/BiP complex boosts PERK signaling by phosphorylating elf2, which activates ATF4 and CHOP, inducing a thorough apoptotic response while limiting the protein translation [197]. A transcription factor downstream of PERK and ATF6 signaling known as CHOP (C/EBP homologous protein) promotes apoptosis during the UPR. OA cartilage has higher CHOP levels [198]. The transcriptional activity of caspase-3 can be improved by upregulating the downstream protein CHOP through the PERK-eIF2a-CHOP axis or ATF6-CHOP axis. Highly expressed caspase-3 can mediate the apoptotic cascade and harm cells [199,200]. According to one study, OA-related alterations in the expression levels of *Col2a1* and *Acan* were repressed in the absence of CHOP, and chondrocyte apoptosis was reduced in CHOP null mice compared to that in the wild-type controls [201].

Several in-depth investigations into the inositol-requiring enzyme 1 (IRE1) signaling pathway have been performed recently, and their findings indicate that this pathway may be connected to the chondrocyte apoptosis [202]. Similar to PERK, IRE1 is capable of deciding cell fate by the following two distinct routes. The first route involves the induction of an adaptive cellular response through the unconventional splicing of the transcription factor X-box binding protein 1 (XBP1) mRNA by activated IRE1 or regulated IRE1-Dependent Decay (RIDD) posttranscriptional modifications. The second route involves the activation of the pro-apoptotic c-Jun N-terminal kinase (JNK) under prolonged or severe stress conditions [203,204]. Moreover, ATF6 is required for XBP1 expression in OA cartilage, and ATF6 can enhance the expression of IRE1α-spliced XBP1S in OA cartilage. Upon induction of ER stress by TNFα and IL-1β, the IRE1α-XBP1 pathway is activated. Then, both the levers of transcription of the XBP1 gene are activated by ATF6, and XBP1S spliced by IRE1α is elevated; XBP1S, in turn, inhibits the ER stress-mediated apoptosis in osteoarthritis. XBP1S is a negative regulator of apoptosis in osteoarthritis [196]. This study revealed that intermediate OA cartilage had increased XBP1 mRNA splicing as a protective ER stress response, but neither mild nor severe OA cartilage had such XBP1 mRNA splicing [198]. In addition to the unusual XBP1 mRNA splicing in intermediate OA, IRE1 degrades a subset of mRNAs that are confined to the ER through the RIDD pathway, promoting cell death [205,206]. Basal RIDD is believed to be the initial homeostatic reaction to ER stress brought on by IRE1. If this reaction is insufficient, RIDD increases and activates XBP1 splicing. However, if ER stress still persists, XBP1 mRNA splicing is suppressed, and RIDD eventually starts the apoptosis [207]. ROS production is known to be a significant manifestation of inflammation during OA development; however, whether intraarticular inflammatory cytokine production can also mediate IRE1-RIDD activation-induced chondrocyte apoptosis is unclear and needs to be further explored [37].

Furthermore, IRE1 interaction with JNK and its related proteins can trigger IRE1-dependent apoptosis. Activated IRE1 interacts with the adaptor protein TNFR-associated factor 2 (TRAF2) via the CD domain during the UPR, resulting in the recruitment of apoptosis signal-regulating kinase 1 (ASK1) and the formation of an apoptosis complex [208,209]. Following the formation of this complex, the JNK pathway is activated, as are downstream molecules, such as Bax and caspase 2 [209]. Circ-0114876 increases TRAF2 expression in a model of IL-1-induced chondrocyte damage, whereas inhibition of circ-0114876 increases chondrocyte activity, attenuates the inflammatory response, and lowers the apoptosis [210]. However, the precise molecular mechanism underlying this circ-0114876 activity is unknown. We might assume that the IRE1-TRAF2-ASK1-JNK signaling pathway regulates chondrocyte death through a pathway that needs to be explored in the future (Figure 2).

## 4. Therapy Options in Osteoarthritis

With the development of medicine, many ways have come up to treat osteoarthritis, with the established treatment for OA being pain control and improving the function of joints [211]. As OA is a chronic disease, the first aim of therapies is to reduce its risk factors. The biggest risk factor for OA is obesity. Some studies have shown that with weight loss comes a lower risk of osteoarthritis [212]. Other studies show that exercising properly can also reduce pain caused by OA. Some specific therapeutic techniques, such as soft tissue mobilization and passive stretching, can help relieve OA symptoms [4]. Certain physical modalities, such as braces, are also useful for OA treatment. Various braces can be used to relieve the pain in various regions caused by OA. For example, unloading knee braces can be used to reduce the symptoms of knee OA; sleeves can treat patellar OA, and thumb base semirigid and rigid splints can be used to treat carpometacarpal osteoarthritis [4]. Alternative therapy for OA is also a major research focus. Pain relief through acupuncture has a significant therapeutic effect on osteoarthritis [213,214]. Among alternative therapies, psychotherapy is very important. Lasers and transcutaneous electrical nerve stimulation can also be used to treat OA [214,215]. In terms of pharmacologic treatment, NSAIDs, acetaminophen, and tramadol are the most common drugs. However, their use is limited due to side effects [4]. In terms of intraarticular treatments, injection of steroids or hyaluronic acid (a kind of viscosupplement) can be used for the treatment of OA [216]; however, the safety of this type of treatment is yet to be confirmed.

Programmed cell death is an important cause of OA, and OA can be treated by targeting various PCD signaling pathways. AMPK/mTOR is an important signaling pathway in autophagy and apoptosis. Some studies have shown that metformin can be used to activate AMPK and simultaneously inhibit mTORC1 from relieving the symptoms and inhibiting the exacerbation of OA. Besides affecting the AMPK/mTOR signaling pathway, metformin can also inhibit the expression of the MMP-13 and MMP-3 proteins. MMP-13 and MMP-3 are enzymes that can induce the degradation of type II collagen to exacerbate joint symptoms of OA [104]. Oral treatment with calcitonin, either directly or as a nano-complex, can be used to decrease MMP-13 expression and CTX-II urinary excretion [217]. Furthermore, some mTOR inhibitors, such as rapamycin, may be injected intra-articularly to treat OA because rapamycin can promote autophagy in chondrocytes at the joints to maintain joint homeostasis [103]. Treatments targeting pyroptosis, an important form of programmed cell death in OA, are also useful. The NLRP3 inflammasome plays a role in the induction of the pyroptosis [75]. Therefore, an inhibitor of NLRP3 could control the exacerbation of OA. For example, the NLRP3 inflammasome inhibitor CY-09 can prevent the exacerbation of the OA [77]. As a ubiquitin-specific protease, USP7 can be activated by H_2_O_2_ and subsequently increase the level of ROS, which, in turn, can increase the NLRP3 inflammasome activation [218]. Therefore, USP7 inhibitors can also inhibit the development of OA. An increase in the levels of inflammatory factors, such as IL-1β and IL-18, induced by the NLRP3 inflammasome, can also worsen OA symptoms; this study showed that combined treatment with disulfiram and glycyrrhizin acid could inhibit the increase in the levels of these inflammatory factors to relieve the symptoms of OA [219]. The P2X7 purinergic receptor is also important for the regulation of OA. As an important activator of the NLRP3 inflammasome, it can induce pyroptosis and produce pro-inflammatory cytokines, such as IL-1β and IL-18. P2X7 is also essential for apoptosis in OA because IL-1β can induce the production of TNF-α, and both these pro-inflammatory cytokines have proapoptotic effects. Owing to their effects on inflammatory reactions, pyroptosis, and apoptosis, P2X7R inhibitors can be used as drugs for OA treatment [79]. As a classical signaling pathway, the NF-κB signaling pathway is also important for development. It can promote the expression of catabolic genes, destructive mediators, and pro-inflammatory factors in osteoarthritis. Therefore, inhibiting the NF-κB signaling pathway could partly inhibit the development of OA. Furthermore, the overexpression of some stimulatory genes can enhance the catabolic effect on chondrocytes, while the overexpression of some inhibitory genes can inhibit it. For example, stimulatory factors, such as IκBζ, SAM68, and KPNA2, can increase chondrocyte damage. Therefore, such stimulatory genes of NF-κB can serve as important therapeutic targets for OA [122]. NF-κB can induce the expression of MMP-1,3, and 13, PGE2, and NO to increase cartilaginous degradation and ECM damage [220]. In addition to knocking out heterogeneous p65 in mature chondrocytes, there are various molecules that can block the NF-κB signaling pathway, such as IKK inhibitors, proteasome inhibitors, and small interfering RNAs [220]. The JAK/STAT signaling pathway can also be a therapeutic target for OA because it is associated with the decrease in collagen II and increase expression of MMPs, which are important for cartilage homeostasis [221,222,223]. Thus, JAK inhibitors are expected to be a new category of therapeutic drugs for OA. The Chinese herbal medicine Artesunate has been validated to prevent the development of OA by inhibiting the JAK/STAT signaling pathway [224]. Moreover, SOCS proteins are potent negative regulatory factors of the JAK/STAT signaling pathway in OA. SOCS can block JAK/STAT signal transduction [225] as well as decrease the inflammatory response in chondrocytes through its effect on IL-10 [225]. Osteoarthritis susceptibility genes can also be a good drug target for osteoarthritis treatment. For example, disease-modifying osteoarthritis drugs (DOMADs), such as intra-articular TGF-β and FGF-18 growth factor therapies and Wnt inhibitors, can act on proteins, the genes of which were highlighted through a GWAS. Histone-modifying proteins and osteoarthritis risk loci are present in a common location. Therefore, histone deacetylase inhibitors can inhibit the expression of MMPs and IL-1 [226] (Table 1).

**Table 1 ijms-24-05364-t001:** The different therapeutic methods of OA.

Therapeutic Method	Typical Molecule or Drug	Target Site	Treatment Outcome	Mechanism	Existing Problems	Reference
Intra-Articular Injection	Corticosteroid, Hyaluronic acid, Platelet-rich plasma	interleukin-1, prostaglandins, leukotriene, MMP9, MMP-11 (Corticosteroid)	Reduce pain and increase joint mobility	Corticosteroid: inhibit the secretion of the molecule that can induce pain. Hyaluronic acid: artificially recover the environment of the joint. Platelet-rich plasma: promote healing of damaged joint	Long-time use damaged cartilage High-risk (HA)	[227,228,229]
Calcitonin treatment	Oral calcitonin, calcitonin-based nanocomplex	Subchondral bone, MMP-13, CTX-II	Reduce the degree of cartilage lesions	Reduce the MMP-13 and CTX-Ⅱurinary excretion	Clinical research is insufficient	[217]
NF-κB signal pathway	Small interfering RNAs, SC-514, KINK-1, PHA-408, Proteosome inhibitors, targeted IκBα ubiquination blockers, Electrophilic compounds, Morroniside	IL-1 receptor, MMP, JNK and MAPK dependent cytokine signaling,	Reduce the inflammatory response	Inhibit the transcription of NF-κB target genes or block the NF-κB signaling	In its infancy in animal models and clinical studies	[217,220]
Caspase-l/IL-1β inflammatory pathway	ICA, licochalcone A	NLRP1 inflammasomes, NLRP3 inflammasomes	Reduce the cartilage damage	Inhibit pyroptosis and extracelluar matrix degradation	Clinical research is insufficient	[15]
JAK/STAT signal pathway	AG490, Artesunate, Acteoside	STAT, SOCS	Delay deterioration	Block the JAK/STAT signaling to inhibit the development of OA	The exact effect and side effect is unknown	[163,169]
PI3K/AKT/ mTOR signal pathway	LY294002, Casodex, rapamycin, 17b-estradiol, FGF18, ghrelin	PI3K, AKT	Reduce the cartilage damage	1. Inhibit the PI3K/AKT signaling to decrease sclerosis in subchondral (LY294002, Casodex, rapamycin) 2. Activate the PI3K/AKT signaling to promote the chondrocyte proliferation (17b-estradiol, FGF18, ghrelin)	Have side effects and the effect is a double-edged sword	[161]
OA genetics and epigenetics therapeutic option	DOMADs, HDAC inhibitor, CRISPR-Cas9	DOMADs: highlighted gene protein in GWAS HDAC inhibitor: MMPs and IL-1	Inhibition of OA development in a mouse model	Improving symptoms by affecting the expression of OA-related genes	Accurate delivery of drugs to the joint tissue and targeted treatment is required	[226]

## 5. Conclusions and Outlook

Cell death plays a vital role in regulating the functions of a healthy human body. As a complex chronic disease, osteoarthritis is greatly affected by programmed cell death. Various kinds of programmed cell death play different roles in osteoarthritis. In this review, we first discussed the relationship between programmed cell death and osteoarthritis. For example, autophagy can inhibit the exacerbation of OA by regulating chondrocyte activities. Iron-mediated ferroptosis can lead to the abnormal death of chondrocytes by increasing the level of ROS and MMP13 and decreasing the level of collagen II. Pyroptosis can use NLRP3 inflammasomes to induce the pyroptosis of chondrocytes. Apoptosis can also induce OA. Moreover, some kinds of receptors can regulate chondrocyte apoptosis. A novel form of cell death mediated by copper ions, called cuproptosis, can also lead to OA. Cuproptosis can block the TCA cycle to cause protein toxic stress and, thereby, chondrocyte death. Although it is unknown whether necroptosis is the cause of cartilage deterioration or its result, necroptosis is also related to OA. The complex Ⅱb made by RIPK1, RIPK3, FAD, and caspase-8 can develop into a necrosome, and PIPK can activate DAMP, which is important for OA development, through a few signaling pathways. It is worth noting that almost all types of PCD can produce different kinds of inflammatory factors, which can induce a severe inflammatory response to exacerbate OA. Further, we showed the role of various molecules, such as the NF-κB, Jak, mTOR, and IER1, in regulating PCD for OA development. NF-κB has a bidirectional role in chondrocyte apoptosis so that it can both promote as well as inhibit chondrocyte apoptosis. mTOR plays an essential role in autophagy, especially through the PI3K/AKT/mTOR signaling pathway. mTORC1 can inhibit autophagy and, thereby, relieve the symptoms of OA. Inflammatory factors can stimulate the AK/STAT pathway to cause the death of chondrocytes.

Finally, we summarize the existing treatments for OA, explaining how most of them only relieve the symptoms of OA and do not treat its causes. Therefore, we believe OA treatments targeting different kinds of programmed cell death and signaling pathways known to regulate the development of OA should be explored. Molecules that can inhibit programmed cell death can be used as therapeutic drugs for OA. Although there are many challenges involved, we firmly believe that thorough experimental and clinical research can allow the use of PCD as a great target for the development of therapeutic drugs for treating OA causes without side effects and other complications.

## Figures and Tables

**Figure 1 ijms-24-05364-f001:**
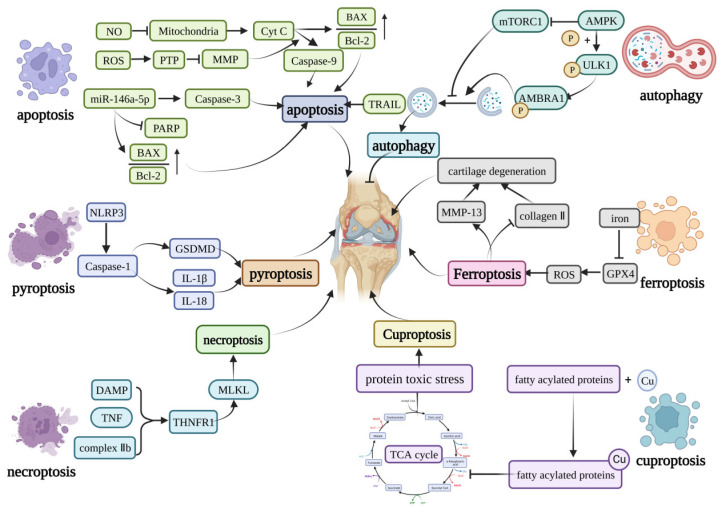
Core molecular mechanisms of programmed cell death in OA. The apoptosis can be induced by caspase-9, caspase-3, TRAIL, and the increase in the BAX/Bcl-2. Pyroptosis is mainly induced by the NLRP3 inflammasome. The NLRP3 inflammasome can activate GSDMD or induce the production of IL-1β and IL-18 to induce pyroptosis. The DAMP, TNF, and complex IIb can induce necroptosis by activating the MLKL. Curprotosis can be induced by the accumulation of copper ions, which can affect the TCA cycle to produce more protein-toxic stress. The over-accumulation of iron can induce ferroptosis by overproduction of ROS, and ferroptosis can lead to the expression of MMP-13 and inhibit the expression of collagen II. The AMPK can inhibit the activation of mTORC1, which can inhibit autophagy or phosphorylation of the ULK1, creating a phosphorylated AMBRA1 to induce autophagy. This figure has been created at https://app.biorender.com (accessed on 21 January 2023).

**Figure 2 ijms-24-05364-f002:**
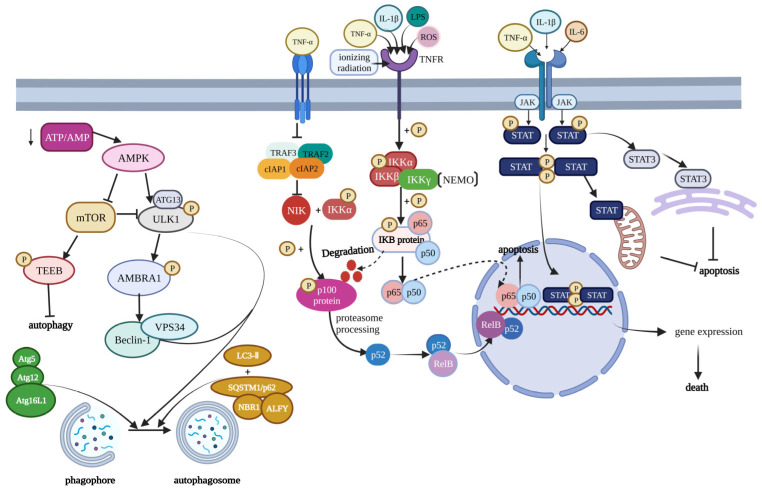
Modulation of cell death pathways in osteoarthritis. This figure contains three main signaling pathways, which play an important role in the regulation of OA. From left to right are the mTOR signaling pathway, NF-κB signaling pathway, and JAK/STAT signaling pathway. The mTOR signaling pathway is mainly influenced by the AMPK, and the decrease of ATP/AMP can activate AMPK. The mTOR can produce the phosphorylated TFEB to inhibit autophagy. Furthermore, the mTOR can also produce the phosphorylated ULK1 complex to inhibit autophagy. Many factors, such as TNF-α, IL-1β, LPS, and ROS, can induce NF-κB signaling. After a cascade of signaling, it will produce the p52 and RelB complex and p65 and p50 complex to influence the gene expression. Furthermore, in the JAK signaling pathway, the JAK can lead to the phosphorylation of STAT. The phosphorylated STAT can form STAT–STAT dimers, which can regulate gene expression, mitochondria, and ER. This figure has been created at https://app.biorender.com (accessed on 21 January 2023).

## Data Availability

Data sharing is not applicable to this article as no datasets were generated or analyzed during the current study.

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
