# Peer review of "The Role of Regulated Programmed Cell Death in Osteoarthritis: From Pathogenesis to Therapy"

_ijms, 2023, doi:10.3390/ijms24065364_

Round 1

Reviewer 1 Report

The authors are trying to summarize the present understanding of the role of programmed cell death in osteoarthritis. Authors try to cover different types of programmed cell deaths and their roles in OA with elaboration over certain pathways like NFkB.

The review does not provide an overall picture of the role of programmed cell death in the disease of osteoarthritis, primarily because of poor writing. The content of the article is acceptable, but it needs extensive reorganization and a better flow to be able to fulfill the purpose of a good review.

Please revise and resend!

All the best!

Author Response

Q1. The review does not provide an overall picture of the role of programmed cell death in the disease of osteoarthritis, primarily because of poor writing. The content of the article is acceptable, but it needs extensive reorganization and a better flow to be able to fulfill the purpose of a good review.

Response: Thank you very much for the reviewer's correction, we have reorganized the article and have cut and blended it to make it more readable (-Line 98-99, 129-132, 134-136,138-144, 167-174, 176-178, 179-180, 181-183, 187-190, 192-193, 198-204, 209-213, 238-240, 251-252, 255-258, 262-268, 270-274, 280-282, 290-292, 352-354, 357-363, 364-380, 389-391, 404-406, 420-422, 447-449, 458-461, 492-494, 619-621). We have made a comprehensive revision of the language. Language polishing certificate attached (Language polishing proof.pdf).

Reviewer 2 Report

The authors provide a comprehensive review article summarizing the ongoing research focusing on the influence of different cell death pathways on the progression of osteoarthritis. The topic is undoubtedly relevant, and it is certainly worthwhile to provide a complete overview regarding the current knowledge obtained in a large number of different studies. There are however different aspects of the manuscript, which need to be improved.

Specific comments:

1)    There are many language problems in the manuscript text, some examples from the Introduction are listed here:

-       “Osteoarthritis is a kind of worldwide chronic disease ...”. The term “a kind of”, which is repeatedly used throughout the manuscript, is certainly not necessary.

-       “All programmed cell death plays an important role in the development of the body, …” is not really a sufficient scientific sentence, and should rather be replaced by something like “Different types of programmed cell death pathways play specific roles in various developmental processes, …”

-       “In the process of apoptosis, the cell contents don´t overflow.” I don´t think that “overflow” is a correct term here.

-       “… some studies have shown that in the cartilage cell, …”. It is probably more appropriate to refer to “chondrocytes” instead of “the cartilage cell”.

Since it is virtually impossible to refer to all of the insufficient sentences throughout the manuscript, I would strongly suggest to take advantage of a professional academic proof-reading service before submitting a revised version of the manuscript.

2)    Another major issue, which is probably more difficult to improve, is the unweighted way of presenting many different studies showing various influences of molecules, pathways and cell death mechanisms that were found to affect chondrocytes and/or osteoarthritis progression.

-       The article structure chosen by the authors is surely acceptable, although it does not fully illustrate that there are many overlapping interactions between the different mechanisms described in sections 2 and 3. One example is Il1-ß, which is repeatedly mentioned to play an important role in most of the cell death mechanisms and signaling pathways presented throughout the article. It might be useful to refer to this issue more specifically and it could be advantageous to illustrate such interactions between different pathways in another Figure.  

-       One major problem essentially explaining the lack of specific drugs to limit osteoarthritis progression is the paucity of knowledge about molecules that have an exclusive expression pattern and function in articular chondrocytes. This is different, for instance, for the treatment of osteoporosis, where molecules like RANKL and Sclerostin have been established as molecular targets for anti-resorptive and osteoanabolic treatment, respectively. In the present article the authors essentially refer to pathways and molecules, which are relevant in many different cell types, and the issue of expected side effects of pharmacological modulators is properly addressed by the authors. Based on these arguments, it would be preferable that the authors introduce some statements about the existence of cartilage-specific molecules and/or osteoarthritis susceptibility genes, which have been found to play a role in the described cell death mechanisms or signaling pathways.

3)    Since this review article already contains a huge amount of information, I would suggest not to overload the principally important Table 1. Here it may be sufficient to focus on the ongoing attempts to improve osteoarthritis therapy, which directly relate to the topic of the present article. Otherwise, the authors would have to additionally explain and to discuss all the other approaches, since, for instance, calcitonin is not mentioned anywhere in the text.

Author Response

Reviewer #2:

Q1. There are many language problems in the manuscript text, some examples from the introduction are listed here.

Response: Thank you very much for pointing out the problem for us. We have carefully read the changes you provided to us and have made linguistic changes based on your suggestions to make our language more scientific and rigorous. We have made a comprehensive revision of the language. Language polishing certificate attached (Language polishing proof.pdf).

Q2. The article structure chosen by the authors is surely acceptable, although it does not fully illustrate that there are many overlapping interactions between the different mechanisms described in sections 2 and 3. One example is IL-1β, which is repeatedly mentioned to play an important role in most of the cell death mechanisms and signaling pathways presented throughout the article. It might be useful to refer to this issue more specifically and it could be advantageous to illustrate such interactions between different pathways in another Figure.

Response: We have made changes based on your suggestions and organized the overlapping and redundant parts of the different mechanisms, such as IL-1b, IL-18 and MMPs, to show their roles more clearly. (-Line 129-132, 134-136,138-144, 167-174, 176-178, 179-180, 181-183, 187-190, 198-204, 209-213, 238-240, 251-252, 255-258, 262-268, 270-274, 280-282, 352-354, 389-391, 404-406, 447-449, 458-461, 492-494, 619-621). We also streamlined the role of NF-κB on osteoarthritis and integrated the relationship between osteoarthritis deterioration and NF-κB. (-Line 357-363, 364-380, 619-621)

Q3. One major problem essentially explaining the lack of specific drugs to limit osteoarthritis progression is the paucity of knowledge about molecules that have an exclusive expression pattern and function in articular chondrocytes. This is different, for instance, for the treatment of osteoporosis, where molecules like RANKL and Sclerostin have been established as molecular targets for anti-resorptive and osteoanabolic treatment, respectively. In the present article the authors essentially refer to pathways and molecules, which are relevant in many different cell types, and the issue of expected side effects of pharmacological modulators is properly addressed by the authors. Based on these arguments, it would be preferable that the authors introduce some statements about the existence of cartilage-specific molecules and/or osteoarthritis susceptibility genes, which have been found to play a role in the described cell death mechanisms or signaling pathways

Response: Based on the reviewer’s recommendations, we reviewed the relevant data to find the susceptibility genes for osteoarthritis and provided treatments for the susceptibility genes (-Line 629-635).

Q4. Since this review article already contains a huge amount of information, I would suggest not to overload the principally important Table 1. Here it may be sufficient to focus on the ongoing attempts to improve osteoarthritis therapy, which directly relate to the topic of the present article. Otherwise, the authors would have to additionally explain and to discuss all the other approaches, since, for instance, calcitonin is not mentioned anywhere in the text.

Response: Due to the redundancy in Table 1, we have removed the first two treatments in the table which are not very relevant to the treatment of osteoarthritis targeting on the programmed cell death (-Line 636). we have also explained more about the table 1 in the text (-Line 589-591)

Round 2

Reviewer 1 Report

Significant improvements were noticed in the revised manuscript. It is much easier to follow what the authors are trying to describe in the review.

Author Response

Reviewer #1:

Thank you very much for your recognition and advice to us.

Reviewer 2 Report

The authors have slightly modified their manuscript by responding to my previous comments. They also took care of the language problem and provide a language polishing certificate.

Although I still think that the wording can be generally improved, there is one sentence, which has to be modified in my opinion, especially since it is the first sentence of the manuscript. In fact, I don´t really understand why the authors use the term "a kind of" to introduce OA in the Abstract.  Why don´t they just say "Osteoarthritis is a chronic disease..." or may be "Osteoarthritis is a highly prevalent chronic disease..."?

Another sentence, which was newly introduced into the revised version (line 711), needs to be modified as well. More specifically, a term like “hetero-KO of p65 in adult chondrocytes” is not acceptable for a scientific publication.

Author Response

Reviewer #2:

Q1. Although I still think that the wording can be generally improved, there is one sentence, which has to be modified in my opinion, especially since it is the first sentence of the manuscript. In fact, I don´t really understand why the authors use the term "a kind of" to introduce OA in the Abstract. Why don’t they just say "Osteoarthritis is a chronic disease..." or may be "Osteoarthritis is a highly prevalent chronic disease..."?

Response: We have revised this sentence according to your suggestion. We have changed the expression of the sentence. (-Line 12)

Q2. Another sentence, which was newly introduced into the revised version (line 711), needs to be modified as well. More specifically, a term like “hetero-KO of p65 in adult chondrocytes” is not acceptable for a scientific publication.

Response: In view of this unscientific expression, we have modified it and made a more scientific expression. (-Line 550, 551). In addition, we conducted a comprehensive review of the article and revised some language issues again.